



# The PROFOUND database for evaluating vegetation models and simulating climate impacts on forests

Christopher P.O. Reyer[1], Ramiro Silveyra Gonzalez[1], Klara Dolos[2], Florian Hartig[3], Ylva Hauf[1], Matthias Noack[4], Petra Lasch-Born[1], Thomas Rötzer[5], Hans Pretzsch[5], Henning Meesenburg[6], Stefan Fleck[6], Markus Wagner[6], Andreas Bolte[7], Tanja GM Sanders[7], Pasi Kolari[8], Annikki Mäkelä[8], Timo Vesala[8], Ivan Mammarella[8,] Jukka Pumpanen[9], Alessio Collalti[10,11], Carlo Trotta[11], Giorgio Matteucci[12], Ettore D'Andrea[12], Lenka Foltýnová[13], Jan Krejza[13], Andreas Ibrom[14], Kim Pilegaard[14], Denis Loustau[15], Jean-Marc Bonnefond[15], Paul Berbigier[15], Delphine Picart[15], Sébastien Lafont[15], Michael Dietze[16], David Cameron[17], Massimo Vieno[18], Hanqin Tian[19], Alicia Palacios-Orueta[20], Victor Cicuendez[20], Laura Recuero[20], Klaus Wiese[20], Matthias Büchner[1], Stefan Lange[1], Jan Volkholz[1], Hyungjun Kim[21], Joanna A. Horemans[22], Friedrich Bohn[23], Jörg Steinkamp[24], Alexander Chikalanov[25], Graham P. Weedon[26], Justin Sheffield[27], Iliusi Vega del Valle[1], Felicitas Suckow[1], Simon Martel[16], Mats Mahnken[1], Martin Gutsch[1], Katja Frieler[1]

[1]Potsdam Institute for Climate Impact Research, Member of the Leibniz Association, P.O. Box 601203, D-14412 Potsdam, Germany
[2]Karlsruhe Institute of Technology (KIT), Karlsruhe, Germany
[3]University of Regensburg, Regensburg, Germany
[4]Fachagentur Nachwachsende Rohstoffe e.V. (FNR), Gülzow-Prüzen, Germany
[5]Technical University of Munich, Munich, Germany
[6]Northwest German Forest Research Institute, Göttingen, Germany
[7]Thünen Institute of Forest Ecosystems, 16225 Eberswalde, Germany
[8]University of Helsinki, Helsinki, Finland
[9]University of Eastern Finland, Kuopio, Finland
[10]National Research Council of Italy, Institute for Agriculture and Forestry Systems in the Mediterranean, Rende (CS), Italy
[11]Department of Innovation in Biological, Agro-food and Forest System, University of Tuscia, 01100 Viterbo, Italy
[12]National Research Council of Italy, Institute for Agriculture and Forestry System in the Mediterranean, Ercolano (NA), Italy
[13]Global Change Research Institute, Brno, Czech Republic
[14]Technical University of Denmark, Lyngby, Denmark
[15]French National Institute for Agricultural Research, Bordeaux, France
[16]Boston University, Boston, USA
[17]Centre for Ecology and Hydrology, Edinburgh, United Kingdom
[18]Centre for Ecology and Hydrology, Lancaster , United Kingdom
[19]Auburn University, Auburn, United States
[20]Technical University of Madrid, Madrid, Spain
[21]University of Tokyo, Tokyo, Japan
[22]Centre of Excellence PLECO, University of Antwerpen, Antwerpen, Belgium
[23]Helmholz Center for Environmental Research, Leipzig, Germany
[24]Senckenberg Biodiversity and Climate Research Centre, Senckenberg, Germany
[25]University of Library Study and Information Technology, Sofia, Bulgaria
[26]Met Office, Wallingford, UK
[27]Princeton University, Dept. Civil & Environ. Eng., Princeton, NJ 08544, USA



Correspondence to: Christopher P.O. Reyer (reyer@pik-potsdam.de)



## Abstract

Process-based vegetation models are widely used to predict local and global ecosystem dynamics and climate change impacts. Due to their complexity, they require careful parameterization and evaluation to ensure that projections are accurate and reliable. The PROFOUND Database (PROFOUND DB) provides a wide range of empirical data to calibrate and evaluate vegetation models that simulate climate impacts at the forest stand scale. A particular advantage of this database is its wide coverage of multiple data sources at different hierarchical and temporal scales, together with environmental driving data as well as the latest climate scenarios. Specifically, the PROFOUND DB provides general site descriptions, soil, climate, $CO_2$, nitrogen deposition, tree and forest stand-level, as well as remote sensing data for nine contrasting forest stands distributed across Europe. Moreover, for a subset of five sites, time series of carbon fluxes, atmospheric heat conduction, and soil water are also available. The climate and nitrogen deposition data contain several datasets for the historic period and a wide range of future climate change scenarios following the Representative Concentration Pathways (RCP2.6, RCP4.5, RCP6.0, RCP8.5). We also provide pre-industrial climate simulations that allow for model runs aimed at disentangling the contribution of climate change to observed forest productivity changes. The PROFOUND DB is available freely as a 'SQLite' relational database or 'ASCII' flat file version (at http://doi.org/10.5880/PIK.2019.008, Reyer et al., 2019). The data policies of the individual, contributing datasets are provided in the metadata of each data file. The PROFOUND DB can also be accessed via the ProfoundData R-package (https://github.com/COST-FP1304-PROFOUND/ProfoundData, Silveyra Gonzalez et al., 2019), which provides basic functions to explore, plot, and extract the data for model set-up, calibration and evaluation.

**Keywords:**

climate change, forest models, model validation, multiple constraints, process-based models, benchmarking, calibration



## 1.   Copyright statement

To be included by Copernicus



## 2. Introduction

Process-based models are key tools for understanding systems and forecasting climate change impacts in ecology and Earth system science (Schellnhuber, 1999). Vegetation is a crucial component of the Earth system, and forests are particularly relevant through their influence on hydrological and biogeochemical cycles, biodiversity and ecosystem services. Process-

based vegetation models are used as diagnostic tools to disentangle the influence of different environmental and human drivers on hydrological and biogeochemical cycling as well as vegetation structure from local, plot-level (Eastaugh et al., 2011; Fontes et al., 2010; Pretzsch et al., 2015; Tiktak and van Grinsven, 1995) to global scales (Chang et al., 2017; Ito et al., 2017). At the same time these models are also the main tools to project climate change impacts on vegetation under changing environmental conditions, again from local (Reyer 2015; Rötzer et al., 2013) to global levels (Zhu et al., 2016).

With increasing model complexity, the inclusion of more and more processes and models being increasingly used to as tools for making quantitative projections for policy and management, there is a strong need to install some quality control on their performance. A basic requirement would be that models are actually able to match observed data. Moreover, while informal methods for calibration and model comparisons were often used in the past, the community has shifted in recent years towards more formal statistical methods for such tasks (Dietze et al., 2013; Hartig et al., 2012), which creates a need for

systematic benchmarking data. For all these tasks, the availability of a wide range of data types crossing different spatial-temporal scales is generally viewed as beneficial (Grimm and Railsback, 2012).

The process of formal calibration, comparison and evaluation of complex vegetation models is often hindered by the availability and the harmonization of suitable data. The data necessary to drive a vegetation model is often complex, and needs to be compiled from different data sources (e.g. Bagnara et al., 2019). In particular for model comparisons, besides

data for the evaluation of individual models, common input and driving data for process-based vegetation models are needed to ensure fair comparisons between the participating models. Although model comparisons have a long tradition in vegetation modeling (Cramer et al., 1999, 2001; Bugmann et al., 1996, Morales et al., 2005), they have often been limited by overall data availability and comparability. Common databases that are ready-to-use for thorough model evaluation would allow the community to gain a better appreciation of model differences, explore structural uncertainties, and provide a basis

for more systematic ensemble projections of climate impacts.

Recently, several initiatives have started compiling model evaluation, input or driving data for a wide range of applications of process-based vegetation models (Huntzinger et al., 2013; Kelley et al., 2013; Warszawski et al., 2014; Sitch et al., 2015). Although these initiatives have leveraged important scientific progress, many of them have focussed on the global scale, mostly providing input and driving data from global products. Such global products generally lack the breadth and depth of

process-level detail required to rigorously assess model performance. The database for the project "Towards robust PROjections of european FOrests UNDer climate change" (hereafter PROFOUND DB) described here, aims to bring together data from a wide range of data sources to evaluate vegetation models and simulate climate impacts at the forest stand scale. It has been designed to fulfil two objectives:



- To allow for a thorough evaluation of complex, process-based vegetation models using multiple data streams covering a range of processes at different temporal scales
- To allow for climate impact assessments by providing the latest climate scenario data.

The PROFOUND DB only provides data for individual forest stands but contains a number of elements that are designed to foster comparison of both global/regional models and local models. The climate data, for example, are provided locally (or bias-corrected using local data) in the same way that stand-scale vegetation models would need them and also extracted from global gridded datasets that global vegetation models would use. The PROFOUND DB is also designed to allow for disentangling of uncertainties that affect quantitative model predictions in ecology (cf. Lindner et al., 2014; Dietze, 2017 for an explanation of different uncertainty types), for example by facilitating standardized evaluations of structural or process uncertainties via model comparisons. Model input and driver uncertainty are addressed through a wide range of climate data from different sources, covering the full range of Representative Concentration Pathways (RCPs). Collalti et al. (2018, 2019) for example, have used the PROFOUND DB to study the effects of thinning on carbon use efficiency across a combination of all four RCPs and five Global Climate Models. Finally, parametric uncertainty can be assessed through the wide range of data that can be used for inverse calibration. In the following we describe the main components of the PROFOUND DB (Reyer et al., 2019) and an R-Package (Silveyra Gonzalez et al., 2019) developed to explore the database and allow rapid and easy access for modellers.



## 3. The PROFOUND database

### 3.1. Forest Site Selection and Concept

The forest sites featured in the PROFOUND DB were selected to provide a wide array of data sources across a European gradient. We focussed in particular on providing long time series of tree- and stand-level growth and yield as well as carbon

5   cycle data available from eddy-flux measurements because these variables are most commonly in calibrating and evaluating process-based vegetation models. The selected sites spread along a wide climatic gradient across Europe (Figure 1, Table 3) and cover some of the most common European forest types, as well as the main central European forest management history of favouring mono-specific, even-aged forests or mixtures of two tree species.

We compiled the data from existing data sources and collected the definitions of variables, their units and information about

10   the main measurement methods from the site principal investigators (PIs) and from official descriptions of the data to harmonize the variables as much as possible. The overall guiding principle for the compilation of the data was to provide data that can be easily used by modellers for setting up and evaluating their models. In order to allow for data uncertainty to be reflected in model calibration studies, we also included uncertainty estimates for the measured data, such as those available for carbon flux measurements (cf. Sect.3.2.9), wherever possible.

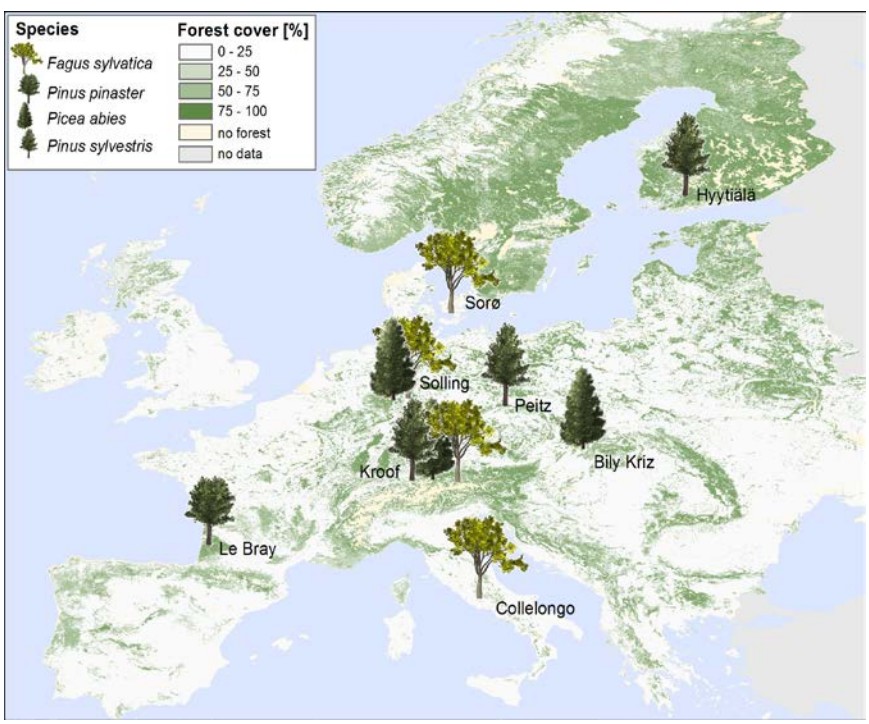

**Figure 1: Location of forest sites and main tree species. Background shows the European forest cover after Brus et al. (2012).**





### 3.2. Data sources

The PROFOUND DB provides information on the site, soil, and forest stand as well as data for climate, atmospheric $CO_2$ concentration, nitrogen deposition, carbon fluxes, atmospheric heat conduction, and remote sensing at a range of different temporal resolutions (i.e., from 30 minutes to decadal measurements). Table 1 provides an overview of the different data types and their temporal resolution available in the PROFOUND DB. All variables available are listed in the SOM tables 1-13. In the following we describe how the individual subdatasets of the PROFOUND DB have been brought together and describe the key variables and characteristics of each dataset.



**Table 1: Overview of the data available in the PROFOUND DB. The years indicate the first and the last year for which data is available except for once-off measurements. The superscript letters indicate the temporal resolution of the data: O = one-off measurement(s), M = 30-min measurements; D = daily measurements; C = 8-day or 16-day composite, A = annual measurements.**

| | Bily Kriz | Collelongo | Hyytiälä | KROOF | Le Bray | Peitz | Solling | Sorø |
|---|---|---|---|---|---|---|---|---|
| Soil | 2011[O] | 1995/2008[O] | 1995/1996[O] | 2003/2004[O] | 1995/2003/ 2004/2005[O] | 2011[O] | 2010[O] | 1997/2004/ 2006[O] |
| Local climate | 2000-2008[D] | 1996-2014[D] | 1996-2014[D] | 1998-2010[D] | 1996-2008[D] | 1901-2010[D] | 1960-2013[D] | 1996-2012[D] |
| Reanalysis climate | 1901-2012[D] | 1901-2012[D] | 1901-2012[D] | 1901-2012[D] | 1901-2012[D] | 1901-2012[D] | 1901-2012[D] | 1901-2012[D] |
| Climate scenarios (ISIMIP2b) | 1661-2299[D] | 1661-2299[D] | 1661-2299[D] | 1661-2299[D] | 1661-2299[D] | 1661-2299[D] | 1661-2299[D] | 1661-2299[D] |
| Climate scenarios (ISIMIPFT) | 1950-2099[D] | 1950-2099[D] | 1950-2099[D] | 1950-2099[D] | 1950-2099[D] | 1950-2099[D] | 1950-2099[D] | 1950-2099[D] |
| Atmospheric $CO_2$ | 1765-2500[A] | 1765-2500[A] | 1765-2500[A] | 1765-2500[A] | 1765-2500[A] | 1765-2500[A] | 1765-2500[A] | 1765-2500[A] |
| Nitrogen deposition (ISIMIP2b) | 1861-2100[A] | 1861-2100[A] | 1861-2100[A] | 1861-2100[A] | 1861-2100[A] | 1861-2100[A] | 1861-2100[A] | 1861-2100[A] |
| Nitrogen deposition (EMEP) | 1980-2014[A] | 1980-2014[A] | 1980-2014[A] | 1980-2014[A] | 1980-2014[A] | 1980-2014[A] | 1980-2014[A] | 1980-2014[A] |
| Forest tree data | 1997-2015[A] | 1992-2012[A] | 2001-2008[A] | 1997-2010[A] | - | 1948-2011[A] | 1967-2014[A] | 1944-2010[A] |
| Forest stand data | 1997-2015[A] | 1992-2012[A] | 1995-2011[A] | 1997-2010[A] | 1986-2009[A] | 1937-2011[A] | 1967-2014[A] | 1944-2013[A] |
| Modis | 2000-2014[C] | 2000-2014[C] | 2000-2014[C] | 2000-2014[C] | 2000-2014[C] | 2000-2014[C] | 2000-2014[C] | 2000-2014[C] |
| Flux | 2000-2008[M] | 1996-2014[M] | 1996-2014[M] | - | 1996-2008[M] | - | - | 1996-2012[M] |
| Meteorological | 2000-2008[M] | 1996-2014[M] | 1996-2014[M] | - | 1996-2008[M] | - | - | 1996-2012[M] |
| Atmospheric heat conduction | 2000-2008[M] | 1996-2014[M] | 1996-2014[M] | - | 1996-2008[M] | - | - | 1996-2012[M] |
| Soil flux series | 2000-2008[M] | 1996-2014[M] | 1996-2014[M] | - | 1996-2008[M] | - | - | 1996-2012[M] |



### 3.2.1. Site information

For each forest site, the PROFOUND DB contains information on general site characteristics such as coordinates, elevation and forest type (Table 2). There is also information on the potential natural vegetation and main tree species belonging to the regional flora (not shown).

5  **Table 2: Overview of the main site characteristics provided for each forest site in the PROFOUND DB.**

| Name | Lat | Lon | Country | Aspect (°) | Elevation (m.a.s.l.) | Slope (%) | FAO Soil type* | Main tree species |
|---|---|---|---|---|---|---|---|---|
| Bily Kriz | 49.30 | 18.32 | CZ | 180 | 875 | 12.5 | Haplic Podzol | *Picea abies* |
| Collelongo | 41.85 | 13.59 | IT | 252 | 1560 | 10 | Dystric Luvisol | *Fagus sylvatica* |
| Hyytiälä | 61.85 | 24.29 | FI | 180 | 185 | 2 | Haplic Podzol | *Pinus sylvestris, Picea abies* |
| KROOF | 48.25 | 11.40 | DE | 1.8 | 502 | 2.1 | Luvisol | *Picea abies, Fagus sylvatica* |
| Le Bray | 44.72 | -0.77 | FR | | 61 | 0 | Arenosol | *Pinus pinaster* |
| Peitz | 51.92 | 14.35 | DE | | 50 | 0 | Dystric Cambisol | *Pinus sylvestris* |
| Solling (beech) | 51.77 | 9.57 | DE | 225 | 504 | 1 | Haplic Cambisol | *Fagus sylvatica* |
| Solling (spruce) | 51.76 | 9.58 | DE | 90 | 508 | 1 | Haplic Cambisol (dystric, densic) | *Picea abies* |
| Sorø | 55.49 | 11.64 | DK | | 40 | 0 | Alfisols/Molisols** | *Fagus sylvatica* |

*according to ISSS-ISRIC-FAO (1998).

**depending on base saturation under or over 50% with a 10-40 cm deep organic layer (cf. Pilegaard et al. 2003)

### 3.2.2. Soil data

The description of the soil profiles contains information about physical and chemical properties of each soil horizon
10  including the organic layer. Unfortunately the soil data are very heterogeneous for the sites and considerable amounts of data are missing. In order not to lose the data that is available for only a subset of sites, we did not harmonize the individual variables but for each site provide the soil data in a consistent format. Despite these limitations, for most sites important soil



data such as the depth of horizons, soil texture, bulk density, field capacity, wilting point, carbon and nitrogen content and pH of the soil solution are available (cf. SOM Table 2).

### 3.2.3. Local climate

For every site we compiled the locally observed daily meteorological data, either from measurement towers or from nearby meteorological stations. These time series cover the main climatic variables required by vegetation models and different time periods for each site (Table 3). They represent the best possible climate information for each site and are most suitable for model simulations comparing simulation output to observations.

### 3.2.4. Reanalysis products

In order to cover longer historical time periods and to assess uncertainties due to the choice of different climate inputs, the PROFOUND DB also provides long historical daily climate time series for each of the sites extracted from four different global reanalysis/observational products:

- Princeton's Global Meteorological Forcing Dataset (PGMFD v.2, hereafter Princeton) from 1901-2012 by Sheffield et al. (2006)
- Global Soil Wetness Project Phase 3 (GSWP3) from 1901-2010 by Kim (Personal Communication, http://hydro.iis.u-tokyo.ac.jp/GSWP3/)
- Water and Global Change programme (WATCH) from 1901-2001 by Weedon et al. (2011)
- WATCH-Forcing-Data-ERA-Interim (WFDEI) from 1901-2010 by Weedon et al. (2014)

Climate variables for the forest stands were extracted from the 0.5° x 0.5° grid cell of the global reanalysis/observational product in which the forest stand is located. The data is then kept at the original 0.5° x 0.5° resolution to allow for comparing the effects of choosing climate inputs for a vegetation model from a global reanalysis product as opposed to the local data presented in Sect. 3.2.3. The difference between the local data and the reanalysis data is most obvious for those sites located in complex, hilly terrain such as Collelongo or KROOF (Table 2). In these hilly locations the grid box average heights of the reanalysis products differ substantially from the heights of the site measurements.

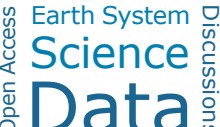

**Table 3. Averages of the daily maximum temperature (Tmax), daily minimum temperature (Tmin), daily mean temperature (Tmean), annual precipitation sum (P), daily mean relative humidity (RH), daily mean air pressure (AP), annual sum of global radiation (R, direct + diffuse shortwave radiation), and daily mean wind speed (W) for each of the sites in the PROFOUND DB from 5 different sources: a locally observed climate and four different global reanalysis/observational products (GSWP3, Princeton, WATCH, WFDEI). The column "Year" indicates the years for which the mean climates have been calculated for the different sources. Please note that the two Solling sites have the same climate.**

| Site | Source | Years | Tmax [°C] | Tmean [°C] | Tmin [°C] | P [mm] | RH [%] | AP [hPa] | R [J cm$^{-2}$] | W [m s$^{-1}$] |
|---|---|---|---|---|---|---|---|---|---|---|
| Bily Kriz | local | 2000-2008 | 11.50 | 7.36 | 3.80 | 1434.56 | 81.99 | 913.19 | 378774.86 | 2.19 |
| | GSWP3 | 2000-2008 | 12.65 | 7.66 | 3.03 | 1034.22 | 76.77 | 957.64 | 395464.73 | 3.71 |
| | Princeton | 2000-2008 | 12.47 | 7.67 | 2.85 | 914.89 | 78.77 | 960.22 | 402658.93 | 3.12 |
| | WATCH | 2000-2001 | 12.72 | 8.25 | 3.43 | 1124.52 | 75.08 | 948.34 | 322865.69 | 2.05 |
| | WFDEI | 2000-2008 | 12.43 | 7.66 | 2.81 | 1034.40 | 76.22 | 950.08 | 438978.13 | 3.25 |
| Collelongo | local | 1996-2014 | 11.46 | 7.24 | 3.46 | 1178.62 | 74.03 | 849.59 | 541888.38 | 1.73 |
| | GSWP3 | 1996-2010 | 20.64 | 15.12 | 10.46 | 977.40 | 68.42 | 903.78 | 530247.74 | 3.83 |
| | Princeton | 1996-2012 | 20.28 | 15.17 | 10.09 | 757.99 | 73.76 | 944.66 | 539045.09 | 4.55 |
| | WATCH | 1996-2001 | 20.57 | 15.21 | 9.99 | 962.33 | 69.66 | 897.07 | 465115.41 | 2.11 |
| | WFDEI | 1996-2010 | 20.40 | 15.12 | 10.22 | 972.10 | 75.02 | 903.20 | 549826.57 | 2.40 |
| Hyytiälä | local | 1996-2014 | 7.40 | 4.36 | 1.13 | 604.01 | 77.95 | 991.08 | 309628.86 | 3.42 |
| | GSWP3 | 1996-2010 | 8.03 | 4.00 | -0.20 | 689.08 | 83.96 | 998.01 | 350511.52 | 3.42 |
| | Princeton | 1996-2012 | 7.88 | 4.06 | -0.37 | 574.87 | 83.41 | 1007.97 | 330041.85 | 3.52 |
| | WATCH | 1996-2001 | 7.93 | 3.88 | -0.17 | 690.02 | 81.29 | 993.85 | 280668.38 | 2.44 |
| | WFDEI | 1996-2010 | 7.97 | 4.00 | -0.26 | 668.75 | 79.23 | 993.60 | 328551.11 | 2.12 |
| KROOF | local | 1998-2010 | 12.99 | 8.15 | 3.91 | 849.46 | 80.73 | NA | 391563.62 | 1.08 |
| | GSWP3 | 1998-2010 | 14.43 | 9.65 | 5.23 | 1014.37 | 80.55 | 954.55 | 423260.65 | 3.04 |
| | Princeton | 1998-2010 | 14.15 | 9.66 | 4.95 | 772.08 | 82.05 | 935.11 | 433277.37 | 3.18 |
| | WATCH | 1998-2001 | 14.48 | 9.83 | 5.39 | 1061.27 | 76.35 | 959.58 | 337605.56 | 2.78 |
| | WFDEI | 1998-2010 | 14.41 | 9.65 | 5.22 | 976.78 | 76.67 | 954.13 | 431629.74 | 2.58 |
| Le Bray | local | 1996-2008 | 17.76 | 13.37 | 9.39 | 920.18 | 76.11 | 1005.81 | 472940.36 | 3.02 |
| | GSWP3 | 1996-2008 | 19.06 | 14.23 | 9.63 | 918.76 | 73.90 | 1014.64 | 490253.28 | 4.90 |
| | Princeton | 1996-2008 | 18.62 | 14.24 | 9.19 | 951.01 | 80.41 | 989.70 | 484739.73 | 4.01 |





| Site | Source | Years | Tmax [°C] | Tmean [°C] | Tmin [°C] | P [mm] | RH [%] | AP [hPa] | R [J cm$^{-2}$] | W [m s$^{-1}$] |
|---|---|---|---|---|---|---|---|---|---|---|
| | WATCH | 1996-2001 | 18.60 | 13.98 | 9.34 | 1095.65 | 74.66 | 1021.76 | 398738.50 | 4.28 |
| | WFDEI | 1996-2008 | 19.20 | 14.23 | 9.78 | 988.57 | 74.37 | 1011.63 | 512514.20 | 2.77 |
| | local | 1901-2010 | 13.50 | 9.02 | 4.93 | 533.10 | 76.37 | 1008.29 | 369794.74 | 2.35 |
| | GSWP3 | 1901-2010 | 13.48 | 9.22 | 5.34 | 654.19 | 75.73 | 1007.39 | 365709.48 | 3.74 |
| | Princeton | 1901-2010 | 13.20 | 9.23 | 5.07 | 557.89 | 85.43 | 999.16 | 374370.83 | 3.51 |
| | WATCH | 1901-2001 | 13.36 | 9.06 | 5.20 | 601.44 | 76.93 | 1007.07 | 309797.89 | 2.79 |
| Peitz | WFDEI | 1901-2010 | 13.47 | 9.18 | 5.23 | 607.58 | 76.54 | 1006.45 | 335821.69 | 3.02 |
| | local | 1960-2013 | 10.54 | 6.75 | 3.39 | 1113.06 | 85.56 | NA | 285026.90 | 1.01 |
| | GSWP3 | 1960-2010 | 11.99 | 8.15 | 4.67 | 933.37 | 79.82 | 988.95 | 355905.60 | 3.95 |
| | Princeton | 1960-2012 | 11.76 | 8.20 | 4.42 | 734.76 | 85.55 | 995.05 | 364950.89 | 3.75 |
| | WATCH | 1960-2001 | 11.65 | 7.79 | 4.38 | 962.00 | 79.38 | 985.97 | 300414.77 | 2.74 |
| Solling | WFDEI | 1960-2010 | 11.89 | 8.14 | 4.58 | 963.98 | 79.21 | 985.95 | 353096.37 | 3.36 |
| | local | 1996-2012 | 10.66 | 8.26 | 5.91 | 760.52 | 82.95 | 1007.71 | 360687.83 | 5.13 |
| | GSWP3 | 1996-2010 | 11.56 | 9.00 | 6.58 | 773.57 | 78.73 | 1012.59 | 376613.02 | 5.86 |
| | Princeton | 1996-2012 | 11.45 | 9.03 | 6.44 | 584.58 | 81.19 | 1005.25 | 363852.90 | 4.98 |
| | WATCH | 1996-2001 | 11.08 | 8.46 | 6.26 | 560.00 | 82.54 | 1009.39 | 343133.71 | 5.66 |
| Sorø | WFDEI | 1996-2010 | 11.52 | 9.01 | 6.55 | 640.02 | 83.06 | 1009.50 | 408098.02 | 4.81 |





### 3.2.5. Climate scenarios

The PROFOUND DB provides climate scenarios based on simulations performed for CMIP5 (https://cmip.llnl.gov/cmip5/) that were bias-corrected and interpolated to a common grid resolution of 0.5° x 0.5° according to Hempel et al. (2013). The climate variables for each site available were extracted from the grid cell of the downscaled climate forcing dataset in which
the forest plot is located. The data can be used in very different ways by the vegetation modelling community:

- The 'ISIMIP Fast Track' scenarios (ISIMIPFT) consist of daily climate data available from five different Global Climate Models (GCMs) (HadGEM2-ES, IPSL-CM5A-LR, MIROC-ESM-CHEM, GFDL-ESM2M and NorESM1-M.) for all four RCPs (Warszawski et al. 2014). The historical period lasts from 1950-2005 and then splits up into the four RCPs from 2006-2099 for each model. The RCPs cover future warming ranges of about 0-9°C in the late
21st century compared to the 1980-2005 average (Figure 2). These ISIMIPFT data are best suited for scenario studies that require a large ensemble of GCMs and RCPs.

- The 'ISIMIP2b' scenarios (ISIMIP2b) consist of daily climate data available from four different GCMs (IPSL-CM5A-LR, GFDL-ESM2M, MIROC5, HadGEM2-ES) for the RCP2.6 and RCP6.0 (Frieler et al. 2017, Lange 2018) as well as RCP4.5 and RCP8.5. The historical period lasts from 1861-2005 and then splits up into the four
RCPs for each GCM from 2006-2099. The RCPs cover future warming ranges of about 1-9°C in the late 21st century compared to the 1980-2005 average (Figure SOM 1). For RCP2.6, RCP4.5 and RCP8.5 from IPSL-CM5A-LR, HadGEM2-ES and MIROC5, additional data are also available for the period 2100-2299. These long-term climatic pathways stabilise at around 1-2°C in the end of the 23rd century compared to 1980-2005 for RCP2.6, around 3-5°C for RCP4.5 and rise up to 16°C for RCP8.5. For all four GCMs, there are also time series of pre-
industrial climatic conditions available from 1661-2299 (or 1661-2099 for GFDL-ESM2M), the so-called pre-industrial control run. The pre-industrial climates from each GCM for the time period 1661-1860 can be combined with the historical climates from 1861-2005 and any future time periods from the corresponding GCM to create a long-term time series of climate data from 1661-2299 (or 2099 depending on the GCM/RCP combinations) without almost any resampling (Frieler et al. 2017). The ISIMIP2b data are best suited to test the implications of long-term
stabilization pathways and different degrees of warming relative to pre-industrial conditions in vegetation models.

- The 'ISIMIP2b locally bias-corrected' scenarios (ISIMIP2bLBC) have the same structure as the ISIMIP2b data but have been bias-corrected using an improvement of the method of Hempel et al. (2013) as described in Frieler et al. (2017) and Lange (2017) and the local observed climatologies presented in Sect. 3.2.3. The ISIMIP2bLBC data are hence best suited for scenario studies that require climatic data to be as consistent as possible with the observational
data (Figure 3).



**Figure 2. Change in mean annual temperature (T mean), annual precipitation sum (P) and annual sum of global radiation (R) over the time period 1950-2099 relative to the 1980-2005 average for the ISIMIPFT scenarios. Please note that the two Solling sites have the same climate.**

**Figure 3. Change in mean annual temperature (T mean), annual precipitation sum (P) and annual sum of global radiation (R) over the time period 1661-2299 relative to the 1980-2005 average for the ISIMIP2b locally bias-corrected (ISIMIP2bLBC) scenarios. Please note that the two Solling sites have the same climate.**

### 3.2.6. Atmospheric $CO_2$ concentrations

Time series of atmospheric $CO_2$ concentrations are provided as annual, global data, hence as one time series for all sites of the PROFOUND DB assuming a well-mixed atmosphere. The historical time series of atmospheric $CO_2$ are based on global atmospheric $CO_2$ concentrations from Meinshausen et al. (2011) from 1765-2005 and have been extended for the period

2006-2015 with data from Dlugokencky & Tan (2014). The future annual atmospheric $CO_2$ concentrations follow the four different Representative Concentration Pathways (RCPs, RCP2.6, RCP4.5, RCP6.0 and RCP8.5) from 2016-2500 from Meinshausen et al. (2011). Figure 4 shows the historical increase in $CO_2$ concentrations since 1765 and the projected future emissions according to the different RCPs. From RCP2.6 till RCP8.5 the total level of $CO_2$ increases strongly and also the date of stabilizing emissions is reached much later in RCP8.5. RCP2.6 is the only RCP that projects declining $CO_2$ levels in

the long run.

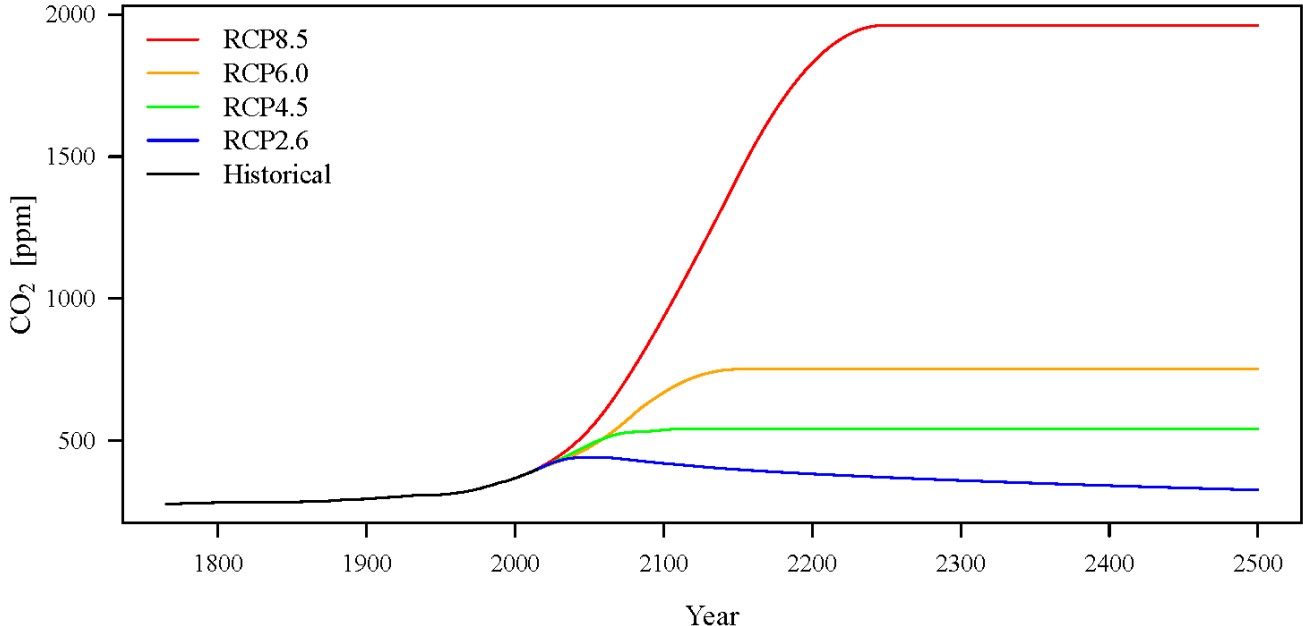

**Figure 4: Global atmospheric $CO_2$ concentrations provided for all sites in the PROFOUND DB. The historical time period extends from 1765-2015 and the scenarios from 2005-2500 for each RCP.**

### 3.2.7. Nitrogen deposition

The nitrogen deposition data, reported as total deposition of reduced and oxidized wet and dry nitrogen deposition, respectively, have been extracted for each site of the PROFOUND DB from two different datasets which serve different purposes.

- EMEP data: For detailed model evaluation studies that require the best possible estimates of local nitrogen deposition, we extracted data from the 'Co-operative programme for monitoring and evaluation of long range





transmission of air pollutants in Europe' (EMEP) for the time period 1980-2014 (EMEP/CEIP 2014). Sea-salt corrected data are available from 1980-1995 in five years steps and from 1986-2014 at annual time step and are derived by atmospheric transport modelling (Simpson et al., 2012).

- ISIMIP data: For model simulation studies, we also provide nitrogen deposition estimates based on atmospheric chemistry modelling for a historical time period (1861-2005) and four future scenarios, where nitrogen deposition follow the four RCPs respectively. The data are further described in Lamarque et al. (2013a, 2013b), sea-salt corrected and consistent with the global nitrogen deposition data provided within ISIMIP (Frieler et al. 2017). The data are taken from the global dataset without further corrections and hence are not intended to represent realistic, local forecasts but rather rough estimates of future nitrogen projections.

For the 1980-2014 time period, the ISIMIP data are typically lower and less dynamic than the EMEP estimates (Figure 5). However, while they do not seem suitable for historical model evaluations, they cover a much longer time period and are clearly interesting for scenario studies because they feature different nitrogen deposition pathways consistent with RCP climates and $CO_2$ pathways. It is also important to note that measured throughfall of $NO_3$ and $NH_4$ is on average lower than modelled total deposition, due to canopy uptake (Marchetto et al in prep.). Moreover, for the two Solling sites the data presented here are identical while in reality total N deposition rates in the spruce stand should be higher because of higher dry depositions. Actually, the ratio between Solling spruce and Solling beech is 1.4 for NH4 throughfall fluxes, 1.6 for NO3 throughfall fluxes, 1.4 for NH4 total deposition, and 1.4 for NO3 total deposition, both using a canopy budget model (Ulrich 1994) for the period 1980-2014. However, these ratios are not constant and are showing an increasing trend over time.



**Figure 5: Total deposition of reduced (NHx) and oxidized (NOy) nitrogen (N) at each of the sites of the PROFOUND DB. The historical period for the EMEP data extends from 1980-2014 and for the historical ISIMIP data from 1861-2005. The future scenarios are available from 2006-2100 and follow the RCP2.6 and RCP6.0 scenarios. Please note that the two Solling sites have the same N depositions (see text for further explanations).**

### 3.2.8. Forest inventory data

For each site, the PROFOUND DB provides information about the forest stand at tree and stand level. The data are available for different time periods and have different measurement intervals, but generally cover mostly the second half of the 20th century and the first decade of the 21st century (Table 1). The data also cover a wide array of height-age and DBH-age relationships (Figure 6-7). For 7 out of 9 sites individual tree diameter at breast height (DBH) and height measurements are available. The time series length ranges between 15 and 65 years within the time period 1948-2015. For the Sorø site, the DBH and heights have been reconstructed from tree-ring data (Babst et al., 2014) and the full stand reconstruction is available from 1944-2010 at annual resolution (cf Text SOM 1). Individual tree data allow analysis and comparison of model simulations with data on single tree growth. From the tree data, we calculated a range of widely used stand variables (cf SOM8). Additional stand-level data are available for some of the sites, such as leaf litter production or leaf area index, and have been included (cf SOM8).

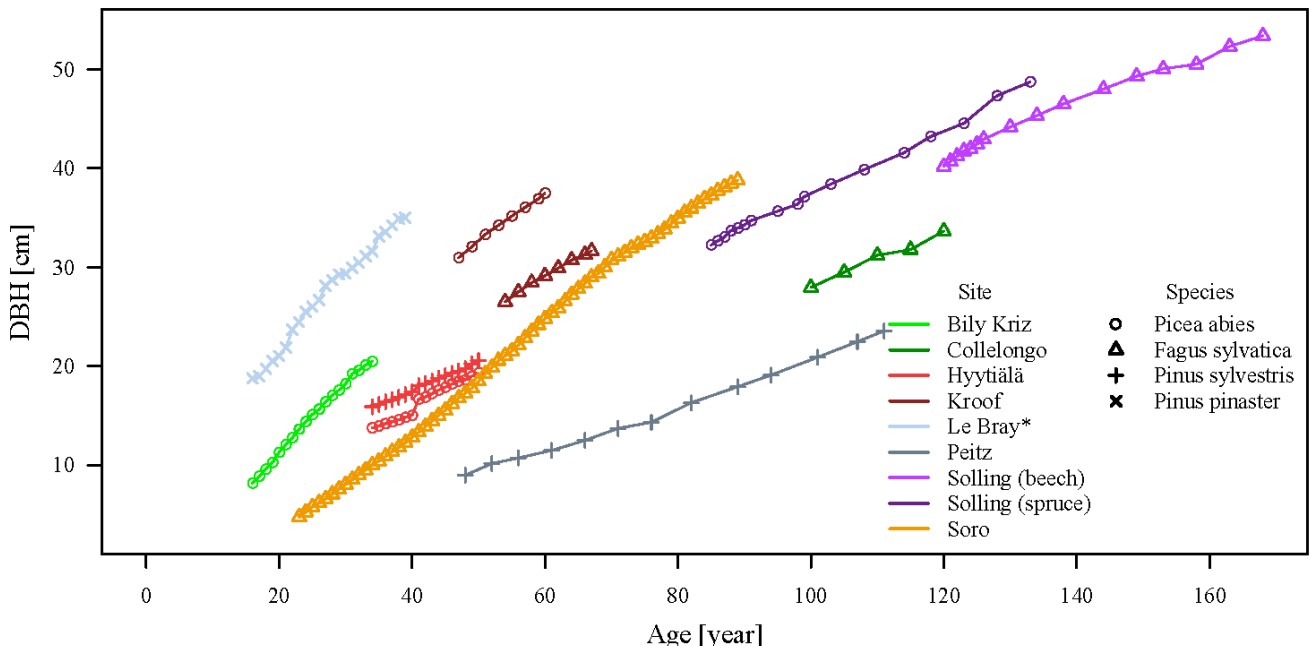

**Figure 6: Time series of tree diameter at breast height (DBH) versus age of the forest stands in the PROFOUND DB. The basal area-weighted mean DBH is shown for all stands with the exception of Le Bray for which the arithmetic mean DBH is shown (marked by \*). For Sorø, the DBHs have been reconstructed (see text in Sect. 4.9 and Text SOM 1).**

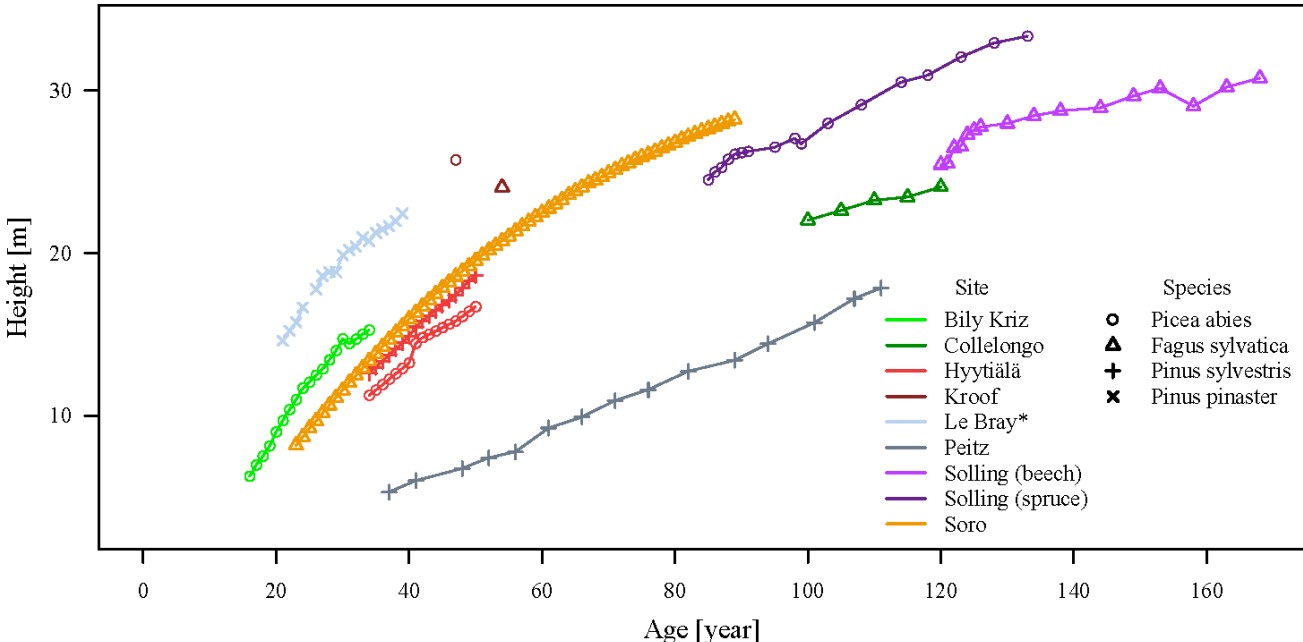

**Figure 7: Time series of tree height versus age of the forest stands in the PROFOUND DB. The basal area-weighted mean height is shown for all stands with the exception of Le Bray for which the arithmetic mean height is shown (marked by \*). For Sorø, the heights have been reconstructed (see text in Sect. 4.9 and Text SOM 1).**





**Table 4. Summary of the main stand variables for the forest stands in the PROFOUND DB. The first number in each cell indicates the value at the first measurement and the second number at the last measurement. The basal area-weighted mean height and DBH are shown for all stands with the exception of Le Bray for which the arithmetic mean height and DBH are shown (marked by \*). The numbers in brackets indicate different data availability for height than for the other variables.**

| Name | Main species | # Obs | Year | DBH [cm] | Height [m] | BA [m²*ha⁻¹] | Age [year] | Stem density [ha⁻¹] |
|---|---|---|---|---|---|---|---|---|
| Bily Kriz | *Picea abies* | 19 | 1997-2015 | 8.16-20.47 | 6.26-15.26 | 10.33-36.96 | 16-34 | 2408-1252 |
| Collelongo | *Fagus sylvatica* | 5 | 1992-2012 | 27.95-33.65 | 22.03-24.08 | 32.25-43.76 | 100-120 | 905-740 |
| Hyytiälä | *Picea abies* | 17 | 1995-2011 | 13.74-19.32 | 11.24-16.7 | 2.96-3.8 | 34-50 | 965-770 |
| Hyytiälä | *Pinus sylvestris* | 17 | 1995-2011 | 15.89-20.58 | 12.61-18.62 | 12.64-18.33 | 34-50 | 870-684 |
| KROOF | *Picea abies* | 8 (1) | 1997–2010 (1997) | 30.96-37.49 | (25.73) | 30.26-39.66 | 47-60 | 512-434 |
| KROOF | *Fagus sylvatica* | 8 (1) | 1997–2010 (1997) | 26.5-31.64 | (24.07) | 12.44-13.2 | 54-67 | 324-220 |
| Le Bray* | *Pinus pinaster* | 24 (18) | 1986-2009 (1991-2009) | 18.76-35.01 | (14.61- 22.44) | 23.3-19.19 | 16-39 | 819-195 |
| Peitz | *Pinus sylvestris* | 13 | 1948-2011 | 8.96-23.54 | 6.75-17.86 | 20.66-36.36 | 48-111 | 4150-886 |
| Solling (beech) | *Fagus sylvatica* | 16 | 1967-2014 | 40.19-53.4 | 25.45-30.78 | 26.99-25.52 | 120-168 | 245-130 |
| Solling (spruce) | *Picea abies* | 17 | 1967-2014 | 32.25-48.74 | 24.51-33.36 | 44-49.46 | 85-133 | 595-290 |
| Sorø | *Fagus sylvatica* | 68 | 1944-2010 | 4.73-38.79 | 8.18-28.23 | 2.15-23.25 | 23-89 | 1767-288 |

5    **3.2.9. Flux data**

The carbon fluxes, i.e. net ecosystem exchange (NEE), ecosystem respiration (RECO) and gross primary production (GPP) are taken from the Tier One Fluxnet2015 dataset (http://fluxnet.fluxdata.org/). We provide estimates of fluxes calculated using different estimates for gap-filled/partitioned fluxes to give a rough estimate of the uncertainty added to the long-term budgets in the process. NEE data are filtered using two different methods to calculate UStar thresholds (Barr et al. 2013, and

10   a modified version of Papale et al. 2006, see also Fluxnet2015 (2017)). Daytime (i.e. Lasslop et al. 2012) and nighttime (i.e. Reichstein et al. 2005) refer to whether ecosystem respiration parameters were estimated from only nighttime fluxes or using





also daytime data (zero intercept of GPP light response curve). In many cases the number of accepted nighttime fluxes is low and the temperature range is narrow, which leads to high uncertainty in the estimated respiration. This can be improved by using also daytime fluxes. On the other hand in the daytime method the uncertainties of photosynthetic light, temperature, and possible VPD responses may be attributed to respiration parameters. Further information about the daytime and

nighttime methods is available in Lasslop et al. (2010) and Reichstein et al. (2005) and also Fluxnet2015 (2017). We also extracted different uncertainty estimates for each variable. Additionally, we provide time series of the sensible and latent heat flux, soil (soil water and soil temperature) and meteorological variables at a 30-min time resolution from the Fluxnet2015 database including measurement uncertainty estimates. Table 5 provides an overview of the main carbon fluxes at each of the sites featured in the PROFOUND DB. Table SOM9 and Table SOM11-13 provides the full list of available

variables.

**Table 5: Summary of the observed carbon fluxes at the sites in the PROFOUND DB. Shown is the range (min & max) and the average (in brackets) of the annual sums in the observational period. All data are estimates based on the CUTRef method with daytime data included for RECO and GPP. GPP is expressed with negative values because it is considered a downward flux from the atmosphere. Likewise negative NEE values indicate a carbon sink and positive**

**a carbon source.**

| Name | Years | NEE [t C ha$^{-1}$] | RECO [t C ha$^{-1}$] | GPP [t C ha$^{-1}$] |
|---|---|---|---|---|
| Bily Kriz | 2000 - 2008 | -91.17 - -32.77 (-65.2) | 54.78 - 102.95 (79.18) | -204.77 - -110.71 (-165.77) |
| Collelongo | 1996 - 2014 | -251.29 - -33.6 (-81.52) | 44.95 - 159.36 (80.79) | -266.75 - -52.59 (-165.46) |
| Hyytiälä | 1996 - 2014 | -81.67 - -12.2 (-24.9) | 16.68 - 115.11 (89.43) | -149.84 - -100 (-117.09)* |
| Le Bray | 1996 - 2008 | -73.96 - 1.04 (-39.15) | 82.36 - 216.09 (145.69) | -236.51 - -126.48 (-194.55)* |
| Sorø | 1996 - 2012 | -82.45 - 8.92 (-19.2) | 151.47 - 223.45 (173.35) | -238.32 - -158.73 (-191.63) |

*year 2007 is without data for Hyytiälä and year 2002 for Le Bray

### 3.2.10.   Remote sensing data

The PROFOUND DB includes remote sensing information at different spatial scales and temporal frequencies, specific for each product. We included five MODIS products (ORNL DAAC 2008a-e) and several vegetation indices calculated from the

surface reflectance data for each of the forest sites. The original MODIS scenes are available at the NASA Land Processes Distributed Archive Center (LP DAAC) (https://lpdaac.usgs.gov/). The specific time series included in the PROFOUND DB were downloaded from the Land Product Subset Web Service of the Oak Ridge National Laboratory Distributed Active Archive Center (ORNL DAAC) (https://daac.ornl.gov/MODIS/). The ORNL DAAC MODIS subsetting Web service is implemented to allow users access to massive amounts of remote sensing data (Santhana-Vannan et al., 2011). In addition a

second set of vegetation indexes were calculated from the reflectance values. A summary of this information is shown in table 6. The full list of variables and how they were aggregated is provided in table SOM10.



**Table 6. Summary of the remote sensing data included in the PROFOUND DB. VIS, NIR and SWIR are the visible, near infrared and shortwave infrared regions of the electromagnetic spectrum.  NDVI: Normalized Difference Vegetation Index, EVI: Enhanced Vegetation Index; FPAR: Fraction of Photosynthetically Absorbed Radiation; LAI: Leaf Area Index;  GPP: Gross Primary productivity; NDWI: Normalized Difference Water Index;  AR: Angle at Red; ANIR; Angle at NIR; AS1: Angle at Shortwave Infrared 1; AS2: Angle at Shortwave Infrared 2; SANI: Shortwave Angle Slope Index; SASI: Shortwave Angle Slope Index**

| Variable | MODIS Source | Spatial Resolution [km] | Temporal Frequency [d] | Time period |
|---|---|---|---|---|
| Reflectance (%) at 7 spectral bands in the optical domain VIS-NIR-SWIR | MOD09A1 | 0.5 | 8 | 2000-2015 |
| Land surface temperature (night & day, Kelvin) | MOD11A2 | 1 | 8 | 2000-2015 |
| NDVI, EVI | MOD13Q1 | 0.25 | 16 | 2000-2015 |
| FPAR·LAI (Dimensionless -1,1) | MOD15A2 | 1 | 8 | 2000-2015 |
| GPP & Net Photosynthesis (gC m$^{-2}$ day$^{-1}$) | MOD17A2 | 1 | 8 | 2000-2014 |
| EVI, NDVI, NDWI (Dimensionless -1,1) | Ratio Indexes calculated from MOD09A1 | 0.5 | 8 | 2000-2015 |
| AR, ANIR, AS1, AS2 (radians, 0-3.14) | Angular indexes calculated from MOD09A1 | 0.5 | 8 | 2000-2015 |
| SANI (-3.14 – 3.14) SASI  (-314-314)) | Angular normalized indexes calculated from MOD09A1 | 0.5 | 8 | 2000-2015 |



The main difference among the forest sites is the data quality, which is highly dependent on the presence of clouds. When possible, low quality observations have been substituted by interpolated values, otherwise the cell was left blank. In any case the alteration of the original data was minimal. It is also important to note that the size of the pixel is large compared to the plot size of the forest stands, which means the pixel data also contain other vegetation than the ones present at the sites.

Three general types of data are included: (1) geophysical variables as measured from the MODIS sensor, i.e. reflectance and temperature, (2) spectral indexes derived directly from reflectance values at different wavelengths, and (3) vegetation properties (i.e. FPAR, LAI, GPP, and Net photosynthesis) as estimated from physical variables through a range of models. Although the MODIS sensor acquires daily information, the PROFOUND DB includes only composite data, that is, for each pixel the best value during a period of time (8 or 16 days) is selected as being representative of that specific period. Spatial

resolution is also specific for each product and is dependent on the physical and technical limitations in the acquisition process of the variables involved in the product computation.

The NDVI and EVI at 250 m spatial resolution coming from the MOD13Q1 product were calculated from the visible and near infrared spectral regions. A temporal frequency (16-day composite) was chosen to minimize the effect of clouds. The EVI Index was developed to correct for atmospheric and background effects so that it shows a larger dynamic range in areas

with high vegetation density (Didan et al., 2015).

The spectral profiles in the whole optical domain (i.e. 459-2155 nm) for each 8-day composite are represented by the surface spectral reflectance at seven wavelengths coming from the MOD09A1 product at 500 m spatial resolution. The criteria for the compositing process are low cloudiness, cloud shadows and low solar zenith angle; when several of these criteria are fulfilled the selection is based on the minimum value in the blue band (Vermote et al., 2015).

The second set of spectral indexes was computed from the MOD09A1 product. The indices based on the spectral shape have the advantage of combining information on three bands instead of two, also when the bands used are located in the SWIR region relevant information related to water is captured (Palacios-Orueta et al., 2005; Khanna et al., 2007; Palacios-Orueta et al., 2012).

LAI is defined as the one−sided green leaf area per unit ground area in broadleaf canopies and as one−half the total needle

surface area per unit ground area in coniferous canopies. The FPAR is the fraction of photosynthetically active radiation (400-700 nm) that is absorbed by the canopy (Myneni, 2015). Gross primary productivity and net photosynthesis estimations are based on the light use efficiency (LUE) concept (Monteith, 1972) using satellite-derived FPAR (from MOD15) and independent estimates of PAR, besides other types of ancillary data. These are highly aggregated variables that have gone through several modelling steps already. Detailed information on the model and information sources used can be found in

Running and Zhao (2015).





## 4. Description of the forest sites

The most northern site is Hyytiälä in Finland with a boreal climate, while the most southern sites are Le Bray in France and Collelongo in Italy with an oceanic and Mediterranean montane climate, respectively. All other sites represent temperate climatic conditions ranging however, from oceanic (Belgium, Denmark), temperate (France, Germany) to sub-continental
(Czech Republic). Unfortunately, sites representing more continental and (east)-Mediterranean forests from southern and south-eastern Europe are missing.

### 4.1. Bily Kriz (CZ)

The Bily Kriz site belongs to the ICP Forests Level II network and is a Fluxnet site located in the Moravian-Silesian Beskydy Mts, Czech Republic, at an altitude of 875 m.a.s.l. The climate is temperate with an annual mean temperature of
7.4°C and an annual precipitation sum of 1434 mm over the 2000-2008 period. The soil is classified as a Haplic Podzol. The site is typical for mountain regions of temperate Europe such as the Black Forest, Bohemian Forest Sumava and forested Carpathians (Hercynian (spruce-)fir-beech forests) but also the higher mountain belts in the (sub-)Mediterranean. Stand forming tree species for such sites are *Fagus sylvatica*, *Abies alba*, and *Picea abies*. Currently, a large part of mixed mountain forests are strongly managed for timber production. The main tree species occurring in Bily Kriz are *Picea abies*
rarely with small proportion of *Fagus sylvatica*. The stand data represent an (even-aged) *Picea abies* monoculture with a mean DBH of 19 cm (year 2015). The potential vegetation belongs to the Geobiocoene type groups: Abieti-fageta (5AB3) - Abies alba Mill. + *Fagus sylvatica* L. with understory: *Calamagrostis arundinacea* (L.) Roth, *Oxalis acetosella* L., *Vaccinium myrtillus* L., *Deschampsia flexuosa* (L.) Trin. More information about the site can be found in Kratochvílová et al. (1989) and Meteorological yearbook (2012).

### 4.2. Collelongo (IT)

The experimental site of Collelongo is located in Selva Piana, a pure *Fagus sylvatica* forest in Collelongo (AQ, central Italy) at 1560 m.a.s.l. Located 100 km from Rome, it is one of the first Italian sites of the ICP network and also part of the ILTER international network. The climate is Mediterranean montane, with a mean annual temperature of 7.2°C and a mean annual precipitation of 1179 mm in the period 1996-2014. Bedrock consists of cretaceous limestone. Soil depth exhibits high spatial
variability ranging from 40 to 100 cm and is classified as a Humic Alisol (Chiti et al. 2010) or Dystric Luvisol according to the FAO classification. The stand is a typical Apennine beech forests dominated by *Fagus sylvatica* with sporadic trees of *Taxus baccata*. The phytosociological association is Polysticho – Fagetum (Feoli & Lagonegro 1982). Currently, Collelongo constitutes a managed *Fagus sylvatica* stand with mean DBH of 25 cm in 2012. In the area around the eddy-flux tower there are only *Fagus sylvatica* trees. Moreover the footprint of the tower is totally included in the *Fagus sylvatica* forest. More
information about the site can be found in Chiti et al. (2010), Collalti et al. (2016) and D'Andrea et al. (2019).



### 4.3. Hyytiälä (FI)

The most northern site included in the PROFOUND DB is the ICP Forests Level II site Hyytiälä, Finland. It is also a Fluxnet site and the coldest site with an annual temperature of 4.4°C and 604 mm annual precipitation during the 1996-2014 period and lies at 185 m.a.s.l. The soil is classified as a Haplic Podzol. *Picea abies* is the naturally dominant tree species building Fennoscandian moss-rich spruce forests with *Pinus sylvestris*. A *Pinus sylvestris* stand was sown in 1962, today with admixtures of *Picea abies* and hardwood species (*Betula pendula*, *Betula pubescens* and *Populus tremula*). Mean DBH were 17 cm for *P. sylvestris*, 5 cm for *P. abies* and 7 cm for hardwood species in the year 2008. More information about the site can be found in Haataja & Vesala (1997), Rannik et al. (2004), Vesala et al. (2005), Ilvesniemi et al. (2009), Mammarella et al. (2009) and Ilvesniemi et al. (2010).

### 4.4. KROOF (DE)

The KROOF forest belongs to the "Kranzberg Forest Roof Experiment" of the Technical University Munich (TUM) and the Helmholtz-Zentrum Munich. The site is located close to Freising, Germany, in the Kranzberger Forst in 502 m.a.s.l (wc-alt.). Mean annual temperature is around 8.2°C, annual rainfall around 849 mm during the period 1998-2010. The soil type, Luvisol, is typical for the region. The potential natural vegetation is (sessile oak-) beech forest (*Fagus sylvatica, Quercus petraea, Quercus robur*). The establishment of the research plot dates back to 1992. The mixed stand comprises large groups of *Fagus sylvatica* surrounded by *Picea abies* with mean DBH of 26 cm and 33 cm in 2010, respectively. Other occurring species are *Acer platanoides* (20 cm), *Pinus sylvestris* (31 cm), *Larix decidua* (26 cm) and *Quercus robur* (29 cm). More information about the site can be found in Pretzsch et al. (1998; 2014) and Matyssek et al. (2014).

### 4.5. Le Bray (FR)

The ICP Forests site Le Bray is located 20 km south-west of Bordeaux, France, at an altitude of 61 m.a.s.l. Mean annual temperature is about 13.4°C and precipitation 920 mm during the 1996-2008 period, constituting a moderate oceanic climate. The soil type is Arenosol (sandy and hydromorphic podzol), which is one of the most common soils in the region. The natural vegetation is formed by deciduous broadleaf forests such as pedunculate oak forests (*Quercus robur*), partly with *Quercus pyrenaica*, *Quercus suber* and *Pinus pinaster*. First measurements were made in 1986 in the monospecific planted *Pinus pinaster* stand. The site experienced a storm in 1999 and lost a large amount of trees. In 2009, the mean DBH was 35 cm. The final clear cut of the site occurred at the beginning of 2009. More information about the site can be found in Porté & Loustau (1998), Bosc et al. (2003) and Berbigier et al. (2001).





### 4.6. Peitz (DE)

Peitz is a long term research plot in eastern Brandenburg, Germany. The site lies at about 50 m.a.s.l. The annual rainfall amounts to more than 608 mm and annual mean temperature is around 9.2°C during the 1901-2010 period. The soil type is a Dystric Cambisol. The potential natural vegetation is a South Scandinavian-east Central European dwarf shrub- and lichen-rich pine forests (*Pinus sylvestris*), partly with *Quercus robur* in the understorey, with *Vaccinium vitis-idaea*, *Calluna vulgaris*, *Cladina spp.*, *Dicranum polysetum* on sandy soils and siliceous rocks. The forest is a pine forest (*Pinus sylvestris*) with a mean DBH of around 23 cm and a stand height of 17 m in 2011. The understorey consists partly of *Quercus robur*. Measurements were started in 1948. More information about this site can be found in Riek & Stähr (2004), Noack (2011; 2012) and about the climate data in Gerstengarbe et al. (2015).

### 4.7. Solling beech (DE)

Solling 304 is a long-term intensive forest monitoring plot (Level II) of the ICP Forests network in central Germany. The plot is also part of the LTER (site LTER_EU_DE_009) and of the permanent soil monitoring programme of the state of Lower Saxony. The site is situated in the center of the Solling plateau at an elevation of about 500 m a.s.l. The mean temperature was around 6.8°C and the mean annual rainfall amounted to 1113 mm during the period 1960-2013. The bedrock consist of Triassic sandstone covered with a 60 to 80 cm deep solifluction layer of loess material from which the soil, classified as an Haplic Cambisol, has developed. The humus type is a typical Moder. The tree layer consists only of European beech (*Fagus sylvatica* L.). *Oxalis acetosella* and *Luzula luzuloides* are the major species of the sparse ground vegetation. Actual vegetation was assigned to the Luzulo-Fagetum typicum and is close to the potential natural vegetation. The forest is a 168-year old stand with a mean DBH of 50 cm and a mean height of 30.7 m in 2016. More information about the site can be found in Meiwes et al. (2009), Meesenburg et al. (2009), Panferov et al. (2009), Le Mellec et al. (2010), Meesenburg et al. (2016) and Fleck et al. (2016).

### 4.8. Solling spruce (DE)

Solling 305 is also a long-term intensive forest monitoring plot of the ICP Forests Level II network in central Germany. As the Solling beech site it belongs to the LTER (site LTER_EU_DE_009) and is a permanent soil monitoring plot of the state of Lower Saxony. It is situated close to the Solling beech site at an elevation of about 508 m a.s.l and has similar site conditions as the Solling beech stand. Potential natural vegetation is a *Luzulo luzuloido Fagetum*. Dominant species of the actual ground vegetation are *Vaccinium myrtillus*, *Polytrichum formosum* and *Dechampsia flexuosa* (Bolte et al. 2004). The forest is a 133-year old Norway spruce (*Picea abies*) stand with a mean DBH of 46.6 cm and a mean height of 33.1 m in 2016. More information about the site can be found in Le Mellec et al. (2010), Bonten et al. (2011), Meesenburg et al. (2016), Fleck et al. (2016) and Wegehenkel et al. (2017).





### 4.9. Sorø (DK)

The ICOS site Sorø (DK-Sor in the FLUXNET and ICOS data bases) is located in Denmark at an elevation of 40 m.a.s.l.. The climate is warm temperate and fully humid with a mean annual temperature of 9°C and annual precipitation sum of 774 mm during the period 1996-2010. The soil has been classified as an Alfisols/Molisols. Potential natural vegetation is

deciduous broad-leaved forest dominated by *Fagus sylvatica*. Other species occurring in the area are *Fraxinus excelsior*, *Larix decidua*, *Picea abies*, *Quercus spp.*, *Acer spp.* However, the region is mostly used as cropland. Data on tree DBH are reconstructed from tree ring measurement (Babst et al. 2014) and historical management information for the time period from 1944 to 2010. The mean DBH of this *Fagus sylvatica* stand was 29 cm in the year 2010. More information about the site can be found in Ladekarl (2001), Pilegaard et al. (2003, 2011), and Wu et al. (2013).

**5.  Forest management of the sites**

The sites available in the PROFOUND DB are managed forests and the historic management can be derived from the tree and stand level data (in terms of reduction of stem numbers). However, for future scenario studies generic, simple management and planting guidelines are available (Table 7-8). This future management corresponds best to "intensive even-aged forestry" as defined by Duncker et al. 2012.

**Table 7 Generic future management scenarios for the main tree species featured in the PROFOUND DB.**

| Species | Thinning regime | Intensity [% of basal area] | Interval [yr] | Stand age for final harvest | References |
|---------|-----------------|------------------------------|---------------|------------------------------|------------|
| *Pinus sylvestris* | below | 20 | 15 | 140 | Pukkala et al. 1998; Fürstenau et al. 2007; González et al. 2005; Lasch et al. 2005 |
| *Picea abies* | below | 30 | 15 | 120 | Pape 1999; Pukkala et al. 1998; Hanewinkel and Pretzsch 2000; Sterba 1987; Lähde et al. 2010 |
| *Fagus sylvatica* | above | 30 | 15 | 140 | Schütz 2006; Mund 2004; Hein and Dhôte 2006; Cescatti and Piutti 1998 |
| *Quercus robur/ petraea* | above | 15 | 15 | 200 | Hein and Dhôte 2006; Fürstenau et al. 2007; Štefančík 2012; Kerr 1996; Gutsch et al. 2011 |
| *Pinus pinaster* | below | 20 | 10 | 45 | Loustau et al. 2005, De Lary 2015, Banos et al. 2016 |



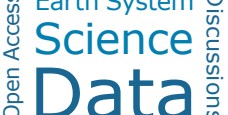

**Table 8 Planting information for the sites included in the PROFOUND DB. The numbers in brackets indicate plausible ranges (na = not available).**

| Name | Density [ha$^{-1}$] | Age [years] | Height [m] | age when DBH is reached [years] | Remarks |
|---|---|---|---|---|---|
| Bily Kriz | 4500 | 4 | 0.5 | 9 | Historical planting density was 5000/ha but current practices are 4500/ha only |
| Collelongo | 10000 | 4 | 1.3 | 4 | Only a rough approximation, usually natural regeneration is the regeneration method. DBH = 0.1 cm at height 1.3m |
| Hyytiälä | 2250 (2000-2500) | 2 | 0.25 (0.2-0.3) | 6 (5-7) | Regenerate as pure pine stand |
| KROOF (beech) | 6000 (5000-7000) | 2 | 0.6 (0.5-0.7) | 5 | The planting density is for single-species stands, hence when regenerating the 2-species-stand KROOF, the planting density of each species should be halved |
| KROOF (spruce) | 2250 (2000-2500) | 2 | 0.35 (0.3-0.4) | 7 | The planting density is for single-species stands, hence when regenerating the 2-species-stand KROOF, the planting density of each species should be halved |
| Le Bray | 1250 (1000-14000) | 1 | 0.2 (0.1-0.25) | 3 (2-5) | These are the current practices (*De Lary, 2015*) and should be used for future regeneration. Historically, the site was seeded with 3000-5000 seedlings per ha and then cleared once or twice to reach a density of 1250 ha-1 at 7-year old when seedlings reach the size for DBH recruitment. |
| Peitz | 9000 (8000-10000) | 2 | 0.175 (0.1-0.25) | 5 | The "age when DBH is reached = 5" is an estimate |
| Solling (beech) | 6000 (5000-7000) | 2 | 0.6 (0.5-0.7) | 5 | The actual stand was established in 1847 from natural regeneration. Until begin of measurements in 1966, the stand was regularly thinned. All figures in table are estimates. Natural regeneration is the recommended regeneration method of stand establishment; stem count in 2014: 130 |
| Solling (spruce) | 2250 (2000-2500) | 2 | 0.35 (0.3-0.4) | 7 | The actual stand was planted in 1891 on a former meadow. Until begin of measurements in 1966, the stand was regularly thinned. All figures in table are estimates.; stem count in 2014: 290 |
| Sorø | 6000 | 4 | 0.82 | 6 | Planted in 1921, stem count in 288 ha$^{-1}$ in 2010, (Wu et al. 2013) |





## 6. The PROFOUND R-package (ProfoundData)

The ProfoundData R-package provides functions to access the PROFOUND DB (Figure SOM2 and SOM3). The ProfoundData package plus a detailed vignette explaining the functionalities are available on Github (https://github.com/COST-FP1304-PROFOUND/ProfoundData). The ProfoundData package serves as interface for users that want to access the PROFOUND DB as a relational database via the R statistical software (R Core Team 2016). The following main functions are included to achieve this goal:

- "getData" to download data (data can be downloaded for one forest site and one underlying dataset at a time)
- "browseData" to check the available forest sites, datasets, variables for a dataset, datasets for a forest site as well as the database version, metadata, data policy, original data source
- "plotData" to quickly inspect any variable of the datasets visually.
- "summarizeData" to summarize data from the database.
- "queryDB" to pass self-defined queries
- "writeSim2netCDF" to write netCDF-files, can be used to convert data (and other files such as model simulation output) into netCDF-files.

While the ProfoundData R-package is meant to provide easy-access to the PROFOUND DB, the database is also fully functional without the R-package.

## 7. Data availability

The PROFOUND Database (http://doi.org/10.5880/PIK.2019.008, Reyer et al. 2019) is available under the Creative Commons Attribution-NonCommercial 4.0 International license (CC BY-NC 4.0). The PROFOUND R-Package (ProfoundData, https://github.com/COST-FP1304-PROFOUND/ProfoundData, Silveyra Gonzalez et al., 2019) is available via a GLP3 license.

## 8. Conclusions

A wide range of data are needed to properly evaluate complex process-based vegetation models. The PROFOUND database compiles data from soil, climate, stand and flux measurements with data from remote sensing, atmospheric nitrogen modelling and climate modelling. Moreover, by providing data at 0.5° x 0.5° grid level plus locally bias-corrected climate data, the datasets can be used to compare local forest models to global vegetation models. The PROFOUND database thus facilitates model evaluation, calibration, uncertainty analysis and model intercomparisons, highlighting the immense value of long term environmental monitoring data for robust inferences about causal processes and future dynamics of forests.





## 9. Appendix: List of Fluxnet sites

**Table A1: List of Fluxnet sites used in PROFOUND DB.**

| Flux sites | FLUXNET-ID | Data-years | Publication | Funding |
|---|---|---|---|---|
| Bily Kriz | CZ-BK1 | 2000-2008 | Kratochvílová et al. (1989), Meteorological yearbook (2012) | Ministry of Education, Youth and Sports of CR within the CzeCOS program, grant number LM2015061 |
| Collelongo | IT-Col | 1996-2014 | Chiti et al. 2010 | EUROFLUX, CARBOEUROFLUX, CARBO EUROPE, CARBO AGE, CARBO EXTREME |
| Hyytiälä | FI-Hyy | 1996-2014 | Haataja & Vesala (1997), Rannik et al. (2004), Vesala et al. (2005), Ilvesniemi et al. (2009), Mammarella et al. (2009) and Ilvesniemi et al. (2010) | ICOS, EUROFLUX, CARBOEUROFLUX, CARBOEUROPE, CARBOEXTREME and by the Academy of Finland Centre of Excellence programme, projects 118615, 141135 and 272041 |
| Le Bray | FR-LBr | 1996-2008 | Porté & Loustau (1998), Bosc et al. (2003) and Berbigier et al. (2001) | INRA, EUROFLUX, CARBOEUROFLUX, CARBO EUROPE, CARBO AGE, CARBO EXTREME |
| Sorø | DK-Sor | 1996-2012 | Ladekarl (2001), Pilegaard et al. (2003, 2011), and Wu et al. (2013) | EUROFLUX, CARBO-EUROPE, CARBO-EUROPE-IP, NITRO-EUROPE, CARBO-EXTREME and Risø-National Laboratory (DK) and technical University of Denmark (DTU) |

## 10. Supplement Link

To be added by Copernicus

## 11. Author Contributions

CPOR and RSG contributed equally to the paper. CPOR & FH initiated the research. CPOR RSG, KD, FH designed the PROFOUND database. CPOR, RSG, YH, KD harmonized and prepared data for the PROFOUND database. RSG programmed the PROFOUND database and R package together with FH, FB and JS. LK and JK provided data for Bily Kriz. AC, GM, CT, EA provided data for Collelongo. PK, AM, TV, IM, JP provided data for Hyyitälä. TR, HP provided data for KROOF. DL, LMB, PB, DP, SL provided data for Le Bray. MN, PLB provided data for Peitz. HM, SF, MW provided data for the Solling sites. AI, KP provided data for Sorø. DC, MV prepared the EMEP Nitrogen data. HT, MB prepared the ISIMIP Nitrogen data. AP, VC, RSG prepared the MODIS data. MB, JV, SL, HK prepared the climate data. SL bias-corrected the climate data. MM and MG checked the data and R-Package. All other authors provided expertise on individual datasets and how to prepare them. CPOR wrote the manuscript with the support of all authors.



## 12. Competing Interests

The authors declare that they have no conflict of interest.

## 13. Acknowledgements

The PROFOUND Database has been developed based on work from COST Action FP1304 PROFOUND (Towards Robust

Projections of European Forests under Climate Change), supported by COST (European Cooperation in Science and

Technology, www.cost.eu), the Intersectoral Impact Model Intercomparison project (ISIMIP) and the I-Maestro project

("Innovative forest management strategies for a resilient bioeconomy under climate change and disturbances, grant no

773324 and 22035418) funded by the ERA-NET Cofund ForestValue and benefited from discussions in the IUFRO Task

Force on Climate Change and Forest Health. We are grateful for the support of all contributing data entities: The Climate

Scenarios have been provided by ISIMIP (BMBF, grant no. 01L1201A1). The initial plot selection was supported with data

from the International Co-operative Programme on Assessment and Monitoring of Air Pollution Effects on Forests (ICP

Forests) operating under the UNECE Convention on Long-range Transboundary Air Pollution (CLRTAP). The data

collection in Bily Kriz was supported by the Ministry of Education, Youth and Sports of CR within the CzeCOS program,

grant number LM2015061. The Collelongo data collection was supported by the projects EUROFLUX,

CARBOEUROFLUX, CARBO EUROPE, CARBO AGE, CARBOEXTREME. The Hyytiälä data collection was supported

by the projects EUROFLUX, CARBOEUROFLUX, CARBOEUROPE, CARBOEXTREME and by the Academy of Finland

Centre of Excellence programme, projects 118615, 141135 and 272041. The KROOF data were provided from TU Munich

funded through the DFG - Sonderforschungsbereich SFB 607 and the DFG - KROOF project „Interactions between Norway

spruce and European beech under drought" (PR 292/12-1, MA 1763/7-1 , MU 831/23-1) as well as by the Bavarian State

Ministry for Nutrition, Agriculture and Forestry and the Bavarian State Ministry for Environment and Health and BaySF

(Bavarian State Forest Enterprise). The data for Le Bray data is kindly provided by INRA funded through the projects

EUROFLUX, CARBOEUROFLUX, CARBO EUROPE, CARBO AGE, CARBO EXTREME.The Peitz data are kindly

provided by Eberswalde Forestry State center of Competence . We are grateful to the Northwest German Forest Research

Institute, Göttingen for providing the Solling Data. Solling Data from 1/2009 to 6/2011 were co-funded LIFE+ by the

Regulation (EC) No.614/2007 of the European Parliament and of the Council, project FutMon (Further Development and

Implementation of an EU-level Forest Monitoring System). The Sorø data collection has been funded through the EU-

projects EUROFLUX, CARBOEUROPE, CARBOEUROPE-IP, NITROEUROPE, CARBOEXTREME and Risø-National

Laboratory (DK) and technical University of Denmark (DTU). This work used eddy covariance data acquired and shared by

the FLUXNET community, including these networks: CarboEuropeIP, CarboItaly and ICOS. The FLUXNET eddy

covariance data processing and harmonization was carried out by the ICOS Ecosystem Thematic Center, AmeriFlux

Management Project and Fluxdata project of FLUXNET, with the support of CDIAC, and the OzFlux, ChinaFlux and

AsiaFlux offices. Graham Weedon was supported by the Joint DECC and Defra Integrated Climate Program – DECC/Defra





(GA01101). CPOR and RSG acknowledge support from the German Federal Office for Agriculture and Food (BLE, grant no. 2816ERA06S). We are also grateful to Kirsten Elger, Katja Henning-Hofmann and Michael Flechsig for their support to make the database open access. We are grateful to the many unmentioned technicians and students for their substantial help to maintain the continuous long-term field observations.



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
