# Peer review of "The PROFOUND database for evaluating vegetation models and simulating climate impacts on European forests"

_Earth System Science Data, 2019_

## Referee Comment (RC1) · Anonymous Referee #1 · 12 Dec 2019

General comments:

Current topic, meeting the need for having comprehensive modelling data sets within structured database (PROFOUND DB) and R-package (ProfoundData), providing opportunity for more simple reproduction of modelling results. Database is particularly valuable in the sense that provide excellent bases for implementation of different modelling approaches using range of environmental variables (soil, climate, forest stand and remote sensing data). R-package is allowing exploration, ploting, extraction of the data, calibration and evaluation of modelling results, which can be useful. Provided long climate time-series together all relevant modelling data are general advantage.

[Figure]

Specific comments:

Manuscript is written quite precisely. It includes large amount of relevant information regarding modelling forest pilot plots. It fully satisfies the form of technical data paper. It is acceptable to remain published in current form.

Technical corrections

Page 18 Line 16,17 – "NO3" and "NH4", numbers should be in subscript, as it is in other text.

Page 20 Line 3,4 - "20th", "21st", st in superscript.

Table 4. - adjust cell values in uniform way

---

## Referee Comment (RC2) · Anonymous Referee #2 · 11 Feb 2020

This database is very well presented and I have only three smallish comments, although the first and second ones are important.

1- There is no mention either in the manuscript title neither in the abstract that this is a database for Europe forests only. I strongly suggest that this comes indicated in the manuscript title and abstract. 2- How does this dataset differentiate from other datasets of its kind for DGVM evaluation, such as ILAMB (Collier et al. 2018 JAMES)? 3- P5L6: hydrological cycle is also a biogeochemical cycle, so why the differentiation?

Apart from that this is a high quality and useful database.

---

## Editor Comment (EC1) · David Carlson (Editor) · 7 May 2020

Please can the authors ensure that they respond thoroughly to reviewer comments?

In particular, one reviewer asks that authors include - and perhaps expand on - the points and text they used in one of their reviewer responses. I quote the text below:

Reviewer question: "How does this dataset differentiate from other datasets of its kind for DGVM evaluation, such as ILAMB (Collier et al. 2018)?"

Author's Reply: "This is an interesting point. Actually, in terms of the scope - providing data for thorough model evaluation ready to be used by modellers alongside a software

packet etc - it is very similar. However, the crucial difference is that our dataset is focussing on the forest stand scale and hence aims at presenting most data at that scale while ILAMB and related databases and tools are global in scope. Additionally, we provide long-term and detailed measurements of forest stand structure which is not available in global datasets."

Assuming authors stand behind such a response, I join the reviewer in asking them to ensure inclusion of this exact or closely related text as part of their introduction and justification.

---

## Author Comment (AC2) · 7 May 2020

Dear Prof. Carlson, Thanks you for your swift and constructive reply which I hope I can clarify easily. We have a section in the introduction that essentially already touched the point of the reviewer (P5L26ff in yesterday's resubmitted manuscript) without citing ILAMB. I had not added a citation of ILAMB there to avoid overreferencing but of course ILAMB is another relevant dataset and following your recommendation, I have adjusted the text in the following way to be very clear and also included this in the updated manuscript:

"Recently, several initiatives have started compiling model evaluation, input or driving

data for a wide range of applications of process-based vegetation models (Huntzinger et al., 2013; Kelley et al., 2013; Warszawski et al., 2014; Sitch et al., 2015; Collier et al 2018). Although these initiatives have leveraged important scientific progress, many of them have focussed on the global scale, mostly providing evaluation, input and driving data from global products. Such global products generally lack the breadth and depth of process-level detail required to rigorously assess model performance at smaller scales as for example they lack long-term and detailed measurements of forest stand structure. The database for the project "Towards robust PROjections of european FOrests UNDer climate change" (hereafter PROFOUND DB) described here, aims to bring together data from a wide range of data sources to evaluate vegetation models and simulate climate impacts at the forest stand scale."

I hope to have stressed our focus on the forest stand scale enough now to clearly differentiate our work from other datasets. Please let me know if you think this section needs more depth in order to make the case clear. I'd be happy to rewrite.

Thanks again for considering our work for ESSD.

All the best Christopher
* * *

---

## Author Comment (AC3) · 7 May 2020

Dear Prof Carlson and editorial staff, I was just uploading my reply to the chief-editor and I expected I would be able to also upload the revised manuscript as last time after submitting my reply but this seems not the case. Please let me know how to upload the revised version including the changes related to ILAMB etc. thanks and bests christopher

———————————————

---

## Author Response (AR1)

Dear Prof. Carlson,

Thank you for handling our manuscript and for your patience with us. We have had several delays in revising the paper 1) because of the current situation with everybody being at home etc. and 2) because we found one error in the tree dataset for the stand Soro. It took us quite some time to get this fixed and we had to add more expertise on reconstructing data from tree rings. This is why we have added a new co-author, Flurin Babst. Other than that, we have made all changes requested by the reviewers and replied to their comments. We have also fixed a few minor issues, spelling mistakes etc. in the database and the paper which are documented in a change log available on the DOI landing page.

Thank you for giving us the opportunity to resubmit our work to ESSD!

On behalf of all co-authors

Christopher Reyer

Reviewer 1:

Current topic, meeting the need for having comprehensive modelling data sets within structured database (PROFOUND DB) and R-package (ProfoundData), providing opportunity for more simple reproduction of modelling results. Database is particularly valuable in the sense that provide excellent bases for implementation of different modelling approaches using range of environmental variables (soil, climate, forest stand and remote sensing data). R-package is allowing exploration, ploting, extraction of the data, calibration and evaluation of modelling results, which can be useful. Provided long climate time-series together all relevant modelling data are general advantage.
Manuscript is written quite precisely. It includes large amount of relevant information regarding modelling forest pilot plots. It fully satisfies the form of technical data paper. It is acceptable to remain published in current form.
**Reply: Thanks for your constructive attitude and for seeing the value of our paper.**

Page 18 Line 16,17 – "NO3" and "NH4", numbers should be in subscript, as it is in other text.

**Reply: Thanks for spotting, changed.**

Page 20 Line 3,4 - "20th", "21st", st in superscript.

**Reply: Thanks for spotting, changed.**

Table 4. - adjust cell values in uniform way

**Reply: Thanks for spotting, changed.**

Reviewer 2:

This database is very well presented and I have only three smallish comments, although the first and second ones are important.

**Reply: Thanks for your constructive attitude and for seeing the value of our paper.**

There is no mention either in the manuscript title neither in the abstract that this is a database for Europe forests only. I strongly suggest that this comes indicated in the manuscript title and abstract.

**Reply: Thanks, very good point: We have changed the title into: "The PROFOUND database for evaluating vegetation models and simulating climate impacts on European forests"**

**And also adjusted the third sentence of the abstract to include a mentioning of Europe:** "The PROFOUND Database (PROFOUND DB) provides a wide range of empirical data on European forests to calibrate and evaluate vegetation models that simulate climate impacts at the forest stand scale."

How does this dataset differentiate from other datasets of its kind for DGVM evaluation, such as ILAMB (Collier et al. 2018 JAMES)?

**Reply: This is an interesting point. Actually, in terms of the scope - providing data for thorough model evaluation ready to be used by modellers alongside a software packet etc - it is very similar. However, the crucial difference is that our dataset is focussing on the forest stand scale and hence aims at presenting most data at that scale while ILAMB and related databases and tools are global in scope. Additionally, we provide long-term and detailed measurements of forest stand structure which is not available in global datasets.**

P5L6: hydrological cycle is also a biogeochemical cycle, so why the differentiation?

**Reply: Good point, we removed "hydrological and"**

Apart from that this is a high quality and useful database.

**Reply: Thanks again for your constructive attitude!**

[revised manuscript text omitted]

Commented [MM2]: The new density in the DB from the new data here is 287 although 288 was measured. Not sure where this 287 now comes from.

[Figure]

**Figure SOM2.** Reconstruction of the tree density at Sorø, division 335. During the years from 1921 to, at least, 1944 there was a canopy of old trees. The blue step function is the reconstructed series using the thinning intervals as in Møller (1933) and thinning intensities that match the observations best (red bars, see Table SOM14).

¶

**Thinning**

Unfortunately thinning of the trees is not anymore documented. Anders Grube, the current forest manager, explained the rule for thinning, which will be applied from now (= 2007 +) on, is thinning interval = tree age / 10 in years. This rule of thumb indicates a slight change from Møller's suggestion, i.e. thinning every 5 years at this stand age. Only a very few examples on how much wood volume was actually thinned can be found. In one example, 2007, thinning was performed in division 336 (planted in 1941, i.e. 66 years) the standing stock was 999 m$^3$ and he removed 40 m$^3$, which is 999/4.11 = 243 m$^3$ ha$^{-1}$ and 40/4.11 = 10 m$^3$ ha$^{-1}$ for standing stock and harvested wood, respectively. This means for this particular site, a relative extraction of 4 %. This is smaller than Møller's suggestion (ca. 11 %, for this age, Møller (1933)). Measurements of tree growth as part of contemporary forest planning (by Klaus Wunsch (KW-plan, c/o) are only performed on some divisions, none of them in the footprint of the tower. For this reason, we decided to use the available data from the forest owner, Sorø Akademi (Table SOM14), and use yield table information and tree rings to reconstruct thinning events and the stand development. We concentrate on division 335 where the inventory, ecological and meteorological measurements have been taken. With this approach it is not possible to reconstruct the exact thinning activity in a certain year, but instead the general forest development is being reconstructed. The thinning events are only accurate within a 3-5 years period.

**Tree-ring data**

In addition to estimating tree density through time, we collected tree-ring data from within the flux-tower footprint in Sorø to reconstruct the growth of individual trees and the stand at annual resolution. This was done in two separate sampling campaigns conducted in 2009 and 2017, which were subsequently merged into one homogeneous dataset. Our sampling targeted two fixed plots, a larger one close to the tower (European beech; 58 trees) and a smaller one at a distance of approximately 100 m (European beech and European ash; 12 trees). For each tree within the two plots, we measured the diameter at breast height, as well as tree and crown base heights using a Vertex IV device (Haglöfs, Sweden). In addition, we recorded the position (distance and azimuth) of each tree relative to the plot center. We then collected two increment cores to the pith of each tree, perpendicular to each other to capture circumferential growth variations. These core samples were brought to the lab and prepared according to standard dendrochronological procedures. We measured annual radial growth increment to a precision of 0.01 mm using a Lintab 5 device and the connected TSAP-Win software. The raw ring-width measurements from each sample were visually and statistically cross-dated to assure that the correct calendar year was assigned to each ring. In addition, we estimated the number of missing rings to the pith of the tree ("pith-offset"), which is required in case these data will be used to calculate tree basal area or biomass increment.

**Stand-scale data from reconstructed single tree data**

After deriving the single tree diameter-at-breast-height (DBH) from the tree-ring data, the individual tree age was calculated. This was done by deriving the tree age at coring from the age of the tree ring and adding 4 years which is assumed to be the time an individual needs to reach breast height (1.3 m) and back projecting the age in time. The individual total tree height was estimated using the site-specific age-dependent height model of Nord-Larsen et al. (2009). In order to derive stand-scale data the reconstructed single tree traits (DBH, height and age) are averaged for seven DBH classes of 10 cm class width. The class assignment of single trees is based on the single tree data in 2009 and kept constant over time. The frequency distribution per DBH class is derived for the period 1994-2017 by interpolation and extrapolation of field measurements in 2005, 2009 and 2017 (table SOM15). This frequency distribution is used to weight the averaged DBH class values in order to calculate stand-scale data of DBH, height and age. Stand-scale tree density is derived as described above.

The full set of reconstructed individual tree data from 1925 to 2017 as well as derived DBH class data and stand data for 1994-2017 can be found in *Soroe_DBH_H_AGE_20200428.RData* (http://doi.org/10.5880/PIK.2020.006/). A description of the data found in this file can be displayed with *cat(Soroe_DBH_H_AGE.l.des)* using the statistical computing software R after loading the data file.

**Table SOM15**: DBH class frequency distribution per year. Note from 2005 to 2017 this is a per annual assessment (n=87) and before that the diameters were extrapolated from 2005/2009 to 1994. DBH classes have 10 cm width.

| year | class 1 | class 2 | class 3 | class 4 | class 5 | class 6 | class 7 |
|------|---------|---------|---------|---------|---------|---------|---------|
| 1994 | 0.069 | 0.241 | 0.115 | 0.322 | 0.195 | 0.057 | NA |
| 1995 | 0.069 | 0.241 | 0.092 | 0.322 | 0.218 | 0.057 | NA |
| 1996 | 0.046 | 0.264 | 0.069 | 0.299 | 0.264 | 0.057 | NA |
| 1997 | 0.034 | 0.264 | 0.08 | 0.276 | 0.287 | 0.057 | NA |
| 1998 | 0.034 | 0.264 | 0.08 | 0.264 | 0.299 | 0.057 | NA |
| 1999 | 0.034 | 0.253 | 0.092 | 0.253 | 0.31 | 0.057 | NA |
| 2000 | 0.034 | 0.241 | 0.092 | 0.264 | 0.31 | 0.057 | NA |
| 2001 | 0.034 | 0.241 | 0.08 | 0.276 | 0.31 | 0.057 | NA |
| 2002 | 0.023 | 0.253 | 0.092 | 0.253 | 0.322 | 0.057 | NA |
| 2003 | 0.023 | 0.253 | 0.092 | 0.207 | 0.368 | 0.046 | 0.011 |
| 2004 | 0.023 | 0.253 | 0.08 | 0.195 | 0.391 | 0.046 | 0.011 |
| 2005 | 0.023 | 0.253 | 0.069 | 0.207 | 0.379 | 0.057 | 0.011 |

| | | | | | | | |
|---|---|---|---|---|---|---|---|
| 2006 | 0.026 | 0.218 | 0.051 | 0.179 | 0.436 | 0.077 | 0.013 |
| 2007 | 0.026 | 0.205 | 0.064 | 0.167 | 0.436 | 0.09 | 0.013 |
| 2008 | 0.026 | 0.205 | 0.064 | 0.154 | 0.449 | 0.077 | 0.026 |
| 2009 | 0.026 | 0.205 | 0.064 | 0.141 | 0.462 | 0.077 | 0.026 |
| 2010 | 0.026 | 0.184 | 0.053 | 0.158 | 0.434 | 0.105 | 0.039 |
| 2011 | 0.03 | 0.136 | 0.045 | 0.152 | 0.455 | 0.136 | 0.045 |
| 2012 | 0.031 | 0.138 | 0.046 | 0.138 | 0.492 | 0.123 | 0.031 |
| 2013 | 0.031 | 0.138 | 0.046 | 0.138 | 0.462 | 0.154 | 0.031 |
| 2014 | 0.019 | 0.151 | 0.057 | 0.094 | 0.358 | 0.283 | 0.038 |
| 2015 | 0.019 | 0.151 | 0.057 | 0.094 | 0.453 | 0.189 | 0.038 |
| 2016 | NA | 0.17 | 0.057 | 0.038 | 0.396 | 0.302 | 0.038 |
| 2017 | 0.019 | 0.151 | 0.038 | 0.057 | 0.377 | 0.321 | 0.038 |